# Identification of miR136, miR155, and miR183 in Vascular Calcification in Human Peripheral Arteries

**DOI:** 10.3390/ijms26199349

**Published:** 2025-09-25

**Authors:** Tom Le Corvec, Mathilde Burgaud, Marja Steenman, Robel A. Tesfaye, Yann Gouëffic, Blandine Maurel, Thibaut Quillard

**Affiliations:** 1Centre Hospitalier Universitaire de Nantes, L’institut du Thorax, Service de Chirurgie Vasculaire, 44000 Nantes, France; tomlecorvec@yahoo.fr (T.L.C.); blandine.maurel@chu-nantes.fr (B.M.); 2Nantes Université, CNRS, INSERM, L’institut du Thorax, 44000 Nantes, France; marja.steenman@inserm.fr; 3Centre Hospitalier Universitaire de Clermont-Ferrand, Hôpital Gabriel Montpied, 63003 Clermont-Ferrand, France; mathilde.burgaud@icloud.com; 4Laboratory of Chromatin Dynamics, Centre de Biologie Intégrative (CBI), MCD Unit (UMR5077), Université de Toulouse, CNRS, UPS, 31062 Toulouse, France; robel.tesfaye@hotmail.com; 5Groupe Hospitalier Paris St. Joseph, Department of Vascular and Endovascular Surgery, 75014 Paris, France; ygoueffic@ghpsj.fr

**Keywords:** atherosclerosis, vascular calcification, miRNAs, vascular smooth muscle cells

## Abstract

Vascular calcification (V) is an independent risk factor for all-cause and cardiovascular mortality. Vascular smooth muscle cells (VSMCs) play a major role in VC as they can acquire mineralizing properties when exposed to osteogenic conditions. Despite its clinical impact, there are still no dedicated therapeutic strategies targeting VC. To address this issue, we used human calcified and non-calcified atherosclerotic arteries (ECLAGEN Biocollection) to screen and identify microRNA (miR) associated with VC. We combined non-biased miRNomic (microfluidic arrays) and transcriptomic analysis to select miR candidates and their putative target genes with expression associated with VC and ossification. We further validated miR functional regulation and function in relation to cell mineralization using primary human VSMCs. Our study identified 12 miRs associated with VC in carotid and femoral arteries. Among those, we showed that miR136, miR155, and miR183 expression were regulated during VSMC mineralization and that overexpression of these miRs promoted VSMC mineralization. Cross-analysis of this miRNomic and a transcriptomic analysis led to the identification of CD73 and Smad3 pathways as putative target genes responsible for mediating the miR155 pro-mineralizing function. These results highlight the potential benefit of miR155 inhibition in limiting VC development in peripheral atherosclerotic arteries.

## 1. Introduction

Vascular calcification (VC) of peripheral arteries is an independent risk factor for all-cause and cardiovascular morbi-mortality [1,2,3]. The risk of mortality can also differ depending on vascular beds affected by VC [4]. Intimal calcification is the main form of VC and consists of calcification development within atherosclerotic lesions. The composition and nature of calcification can affect plaque stability. While microcalcifications in proximity to the lumen are linked to an elevated risk of plaque rupture and acute events, recent research indicates that in the context of coronary arteries, intimal macrocalcifications may serve to locally stabilize vulnerable lesions [5,6]. Furthermore, arterial stiffness resulting from VC represents an additional risk factor for morbi-mortality, as it contributes to the development of hypertension and heart failure. Additionally, the protrusion of calcified nodules within the lumen can directly facilitate thrombosis [7]. Ultimately, VC has a major direct negative impact on clinical practice, jeopardizing surgical or endovascular treatments and directly affecting the success of interventions and prognosis in patients with peripheral artery diseases (PADs) [8,9,10].

The development of VC is the result of an imbalance in pro-calcifying factors in pathological conditions such as atherosclerosis and chronic kidney disease. Different mechanisms lead to VC, including high phosphorus concentrations, cell death, and inflammation, which can prompt vascular smooth muscle cells (VSMCs) to transdifferentiate and acquire osteoblastic properties and overexpress genes like *Runx2*, *Sox9*, *Osteopontin*, *Osteocalcin*, and *Alkaline Phosphatase* [11,12]. Conversely, VC inhibitors like pyrophosphate are decreased under these pathological conditions, due to lower production of Ecto-nucleotide Pyrophosphatase/Phosphodiesterase 1 (ENPP-1), and/or their conversion into inorganic phosphate by Alkaline Phosphatase is increased [13].

The post-transcriptional regulation of gene expression by microRNAs (miRs) has demonstrated its significance in human diseases and vascular calcification of coronary arteries [14,15]. Some miRs have a protective effect in inhibiting VSMC osteoblastic transdifferentiation (miR34-b, -205, and -26-a) [16,17,18] and by regulating the Beta Glycerophosphate/Runx2 pathway (miR204 and -140-5p) [19,20]. Conversely, overexpression of miR17-5p, -29, and -221/222 affects regulatory pathways that promote VC [21,22,23].

Previous studies conducted by our research team using human arteries (ECLA and ECLAGEN biocollections) have demonstrated that arteries develop heterogeneous lesions and mineralization between vascular beds [11,24,25]. Histological analysis showed mostly amorphous calcification in carotid lesions, while plaques in femoral arteries exhibit a higher calcium content and are also more prone to bone-like calcification and osteoid metaplasia [24].

Identifying the miRs that regulate vascular calcification (VC) in peripheral arteries is important for improving our understanding and exploring new therapeutic strategies. With this in mind, we used our human biocollection samples of peripheral arteries to identify the miRs associated with all calcification forms in these locations.

In addition to the miRs of interest associated with intimal calcification and ossification, we considered those highly expressed in healthy femoral arteries. These arteries exhibit a molecular signature indicating a higher propensity for mineralization, as confirmed by in vitro mineralization assays on primary VSMCs isolated from healthy femoral arteries compared to other vascular beds [11,26]. We combined this miRNomic analysis with our transcriptomic data to identify the target genes through which the miRs might regulate these processes.

The aim of this study was to identify, in an unbiased manner, the miRs associated with VC in human carotid and femoral arteries, to characterize their involvement in VSMC mineralization, and to determine their target genes and functional role.

## 2. Results

### 2.1. miRs Associated with Vascular Calcification and Ossification in Human Atherosclerotic Lesions

To identify the miRs associated with VC in human plaques, we performed dedicated miR microfluidic arrays that included 753 miRs on 60 arterial samples from the ECLAGEN biocollection. A total of 20 atherosclerotic carotid arteries (10 calcified, 10 none calcified), 20 atherosclerotic femoral arteries (10 ossified, 10 none ossified), 10 healthy carotid arteries, and 10 healthy femoral arteries were examined. We identified calcification and ossification based on the presence of histological calcified structures (clear centers, sheet-like structure, nodules, and osteoid metaplasia) as previously described and validated with microCT [11].

To select our miRs of interest associated with vascular calcification and ossification, we first compared calcified versus non-calcified carotid plaques and showed significant differences in expression for miR515-3p, -518e, -383, -136, -497, -548L, and -183 (Figure 1A). We also identified miR48c, -1278, -640, -155, -127-5p, -554, -183, and -573 when comparing femoral plaques with osteoid metaplasia (bone-like structures) versus no osteoid metaplasia (Figure 1B). Among these targets, we further studied the most regulated miRs (miR515-3p, -518e, -383, -136, -497, -548L, -548c, -1278, -640, -554, -127-5p, and -155) and miR183, which was the only miR associated with both calcification and ossification.

Most of them were differentially expressed in femoral arteries compared to carotid arteries (Appendix A), as we previously showed that femoral arteries are more prone to calcification and ossification compared to carotid arteries [11], and also differentially expressed between healthy and atherosclerotic arteries (Appendix A).

In silico analysis of the biological functions associated with the combined target genes of the 12 miR candidates showed a predominant role in smooth muscle cell lineage differentiation, including Wnt signaling and bone resorption, confirming their putative role in VSMC transition to an osteogenic phenotype (Appendix A).

### 2.2. Selection of the Most Promising miRs Regulated During VSMC Mineralization

VSMCs in lesions are believed to be the main drivers of VC because these cells can acquire osteoblastic properties. Our strategy focused on potential miRs directly implicated in VSMC mineralization, considering that some of the miRs identified in entire lesions could also be indirectly related to calcification. To select the best candidates most likely to regulate VSMC mineralization, we analyzed the relative expression levels of the 12 miRs of interest in primary arterial VSMCs to identify those expressed in VSMCs and regulated during mineralization.

#### 2.2.1. VSMC Mineralization

VSMCs are important drivers of VC, as their phenotypic plasticity can lead to osteogenic properties. To identify miRs involved in this process, we investigated their expression in primary VSMCs and their regulation during cell mineralization induced either by inorganic phosphate or by a cocktail used to induce osteoblastic differentiation from mesenchymal progenitors (beta-glycerol phosphate, dexamethasone, and ascorbic acid). We first validated the mineralization of primary human VSMCs with both protocols and compared their phenotypic modulation in response to these protocols.

For inorganic phosphate-enriched medium, cell mineralization after 7 days was associated with overexpression of *Alkaline Phosphatase* (2.766-fold at day 1), *Osterix* (2.661-fold at day 2), *Collagen I-α1* (1.511-fold at day 2), *Sox9* (2.251-fold at day 5), *Osteocalcin* (2.990-fold at day 3), and *Bone Sialoprotein 2* (4.724-fold at day 1) in the early stage. Conversely, a decrease in contractility-associated gene expression of *α-SMA* (0.313 at day 7), *Calponin-1* (0.453 at day 7), *Tropomyosin-1* (0.692 at day 2), and *Transgelin* (0.420 at day 3) further illustrated the transition of VSMCs into an osteogenic phenotype (Appendix A).

For osteogenic medium (OB), mineralization obtained after 21 days was also correlated with overexpression of *Alkaline Phosphatase* (3.183-fold at day 10), *Osteopontin* (1.873-fold at day 10), and *Osteocalcin* (3.064-fold at day 7). We also found a tendency for overexpression of *CollagenI-α1*. In contrast to inorganic phosphate, the osteogenic medium increased *Runx2* expression throughout the treatment (8.651-fold at day 21) (Appendix A).

#### 2.2.2. Selection of the miR Candidates According to Their Expression and Regulation During VSMC Mineralization

During Pi-driven mineralization, we observed regulation for three miR candidates (miR136, -155, and -183). Increased expression of miR136 (1.828-fold at day 2), miR155 (2.227-fold at day 7), and miR183 (4.53-fold at day 5) was apparent (Figure 2A). The OB pro-mineralizing medium also transiently regulated four miRs of interest. We observed an up-regulation of miR136 (2.346-fold at day 7), miR155 (5.242-fold at day 7), and miR183 (3.45-fold at day 7). Conversely, we observed a down-regulation of miR127 (0.357 at day 21) (Figure 2B). Among the 12 mRs of interest screened, miR383, miR497, miR515, miR518, miR548, miR554, miR640, and miR1278 were not retained as miR candidates, as their relative expression levels were undetectable by qPCR during mineralization induced by both pro-mineralizing conditions.

#### 2.2.3. Functional Analysis After miR Candidate Transfection

To assess the functional importance of miR136, miR155, and miR183 in VSMC mineralization, we carried out miR mimic transfection by lipofection. Using this approach, miR overexpression was maintained for at least 7 days after transfection (Figure 3A).

The analysis of miR overexpression effects during VSMC mineralization (Figure 3B) showed a significant increase in mineralization with miR155 compared to the control (59.6 vs. 37.8%, *p* = 0.0007). We observed a minor increase in mineralization after transfection of mimics miR136 and miR183. Transfection of mimic miR127 did not result in any increase in cell mineralization over mimic control, in addition to a lack of up-regulated expression, such that it was excluded as an miR candidate in the following functional analysis.

Overexpression of miR155 in phosphate-enriched mineralization medium led to a significant increase in the expression of several osteoblastic genes: *Runx-2* (1.353-fold, *p* = 0.014), *Sox-9* (1.914-fold, *p* = 0.014), *Osteopontin* (1.361-fold, *p* = 0.008), *Osteocalcin* (1.532-fold, *p* = 0.0006), and *Bone Sialoprotein-2* (2.551-fold, *p* = 0.027). We also observed a significant decrease in the expression of the VSMC-specific markers *α-SMA* (0.195, *p* = 0.029), *Calponin* (0.154, *p* = 0.029), *Transgelin* (0.359, *p* = 0.029), and *Tropomyosin-1* (0.561, *p* = 0.029) (Figure 3C), which was confirmed at the protein level by Western blotting (Figure 3D).

The overexpression of miR136 also resulted in a significant increase in the expression of osteoblastic genes: *Runx-2* (2.181-fold, *p* = 0.001), *Alkaline Phosphatase* (2.391-fold, *p* = 0.022), and *Bone Sialoprotein-2* (2.159-fold, *p* = 0.029), but we did not observe any significant decrease in contractile VSMC gene expression (Figure 3C).

The overexpression of miR183 also resulted in a minor but significant increase in the expression of the osteoblastic genes: *Osteocalcin* (1.876-fold, *p* = 0.0001) and *Sox-9* (1.608-fold, *p* = 0.001). Conversely, there was a significant decrease in *Collagen-1* (0.647, *p* = 0.0001) and the VSMC markers *α-SMA* (0.519, *p* = 0.029) and *Calponin* (0.455, *p* = 0.029) (Figure 3C,D).

Among these miR candidates, miR155 appeared to play the most effective role in VSMC mineralization and osteogenic transdifferentiation.

### 2.3. miR155 Targeted Gene Identification

In order to identify the genes targeted by the microRNAs (miRs) implicated in vascular calcification in human lesions, a comprehensive list of direct target genes for these miRs was generated. This list was derived from the miRWalk 2.0, miRDB, and miRBase databases. Putative direct target genes of our miRs of interest regulated in calcified and ossified lesions are illustrated in Figure 4A,B. A gene ontology analysis of their functional annotations highlighted their contributing roles in biomineralization (Figure 4C).

In addition, the transcriptomic analysis performed on the biocollection was utilized to identify genes related to tissue mineralization. These genes were selected based on their regulation, which was consistent with the expression of the respective miRs in the corresponding calcified samples (Appendix A). For miR155, our most promising candidate, this strategy led to the identification of two genes: *Smad3* and *CD73* (Figure 5A), which exhibited lower expression levels in calcified and ossified arteries. Consistently with our projection, *Smad3* can play an inhibitory role in phosphate-induced vascular smooth muscle cell calcification [27] and *CD73* can down-regulate the activity of Alkaline Phosphatase, a major enzyme that regulates the extracellular concentration of Pyrophosphate (a potent inhibitor of VC) [28].

To confirm the link between miR155 and the genes that may control its pro-mineralizing properties, we examined the transcriptional and protein expression regulation of CD73 and Smad3 in VSMCs after transfection of miR155 in the control medium (Figure 5B). The transcriptional study showed a significant decrease in *CD73* and *Smad3* gene expression induced by miR155 compared to the control (0.630, *p* < 0.0001, and 0.770, *p* < 0.0001, respectively). This was further confirmed at the protein level for CD73 (0.754, *p* = 0.0002).

## 3. Discussion

In the present study, we identified miRs that are associated with VC in human atherosclerotic lesions and that participate in VSMC mineralization in vitro. Other studies have investigated the expression profiles of miRs in human atherosclerotic plaques and VC. Raitoharju et al., based on the Tampere Vascular Study, compared miR expression in atherosclerotic plaques from carotid, femoral, and aortic arteries. The atherosclerotic plaques were compared with samples from the territory of the internal mammary artery, which is known for its low susceptibility to atherosclerosis. This work showed a significant up-regulation of the expression of miR21, -34a, -146a, -146b-5p, and -210 (4.61-, 2.55-, 2.87-, 2.82-, and 3.92-fold changes, respectively) [29]. Some of them have been associated in vitro with VSMC mineralization [30]. In parallel, recent studies have identified miRs that regulate vascular calcification in different models, both in vivo and in vitro. For example, miR16-5p, miR17-5p, miR20a-5p, miR106b-5p, and miR204/miR-211 have been described as protective against medial calcification and are depleted in extracellular vesicles in chronic kidney disease patients [31,32]. Similarly, miR126-3p, miR33a-5p, and miR223-3p have been shown to reduce calcification in aortic valves in murine experimental models [33,34]. In addition to these studies, emerging data highlight the predictive value of miRs in cardiovascular disease. For VC, serum miR-125b levels associated with VC severity could be used to reflect uremia-associated calcification in chronic kidney disease patients [35].

Our study focused on intimal calcification, a very common and heterogeneous form of VC. We employed a non-biased strategy to identify miRs associated with VC in carotid and femoral arteries. By considering both carotid calcifications and femoral ossifications, our initial objective was to identify miRs that may be involved in these two major types of mineralization and territories. We assumed that common factors might still emerge (like miR183). However, only a few common miRs were identified. This strategy allowed us to identify a dozen miRs enriched or depleted in calcified or ossified lesions, so potentially important in VC, but maybe at different stages of their development. After a validation step in vitro, we showed that most of these miRs were not expressed in primary human VSMCs, indicating that they may also act indirectly in VC, potentially by regulating endothelial, immune, and/or fibroblast cell biology, which could affect VC. They could also be expressed by more differentiated local osteoblastic cells that we did not replicate in vitro with mineralizing VSMCs. The cellular origin of these miRs and their functional relevance in VC will be investigated in future studies.

This strategy also aimed for the first time to identify miRs involved in various forms of VC by examining both the carotid and femoral arteries, which can host different types of VC [11]. We also assessed miR expression during VSMC mineralization using two protocols widely used in the literature. Inorganic phosphate and an osteogenic cocktail can both trigger VSMC mineralization with specific kinetics and molecular pathways [36]. One is commonly used as a model for VSMC mineralization, as seen in patients with chronic kidney disease, associated with hyperphosphatemia and medial calcification, and the latter is employed to mimic osteoblast differentiation from mesenchymal progenitors. In this study, we selected miRs that were commonly regulated during mineralization with each protocol, suggesting that they could act as shared molecular players in these processes.

To date, there are still no dedicated pharmacological therapeutic strategies targeting VC. In our study, miR155 had the most pronounced pro-mineralizing property compared to miR136 and miR183. This confirms recent studies reporting the involvement of miR155 in VC in/ex vivo. While we could not validate miR155 inhibition with anti-miR transient transfection experiments, using murine aorta rings, miR155 deficiency attenuated calcification, and miR155^-/-^ mice showed reduced VC induced by vitamin D3 [37]. Moreover, Zhao et al. found an important regulatory role for the RCN2/STAT3/miR155 loop in hyperphosphatemia-induced VC, particularly in patients with chronic renal failure [38]. Finally, the study by Fakhry et al. also reported the overexpression of miR155 in phosphate-induced rat aortic mineralization [39]. The study by Kordaß et al. identified miR155 as one of the main inhibitors of the CD73 target in the fields of oncology and immunology [40]. Furthermore, we observed a decrease in miR127 expression during mineralization induced by an osteogenic milieu containing dexamethasone, vitamin D3, β-glycerophosphate, and ascorbic acid. Although miR127 overexpression did not affect Pi-induced mineralization, miR127 might suppress VSMC osteoblastic differentiation because specific mechanisms can regulate each of these mineralization models [36]. As miR127 was also shown to increase CD73 expression, this further confirmed the key role of CD73 in VSMC mineralization and tissue calcification [40]. Additional exploratory and mechanistic studies will be required to establish the precise direct and/or indirect link between miR155 and miR127 and CD73 and Smad3 in human calcified tissues.

To our knowledge, this study is the first to aim to identify miRs associated with VC in a cohort of human atherosclerotic peripheral lesions with its putative targets using cross-analysis with transcriptomic data. Our results highlight CD73 and Smad3 as likely direct or indirect targets that contribute to VC both in calcified peripheral arteries and in human primary mineralizing SMCs. In addition to VC, miR155 plays a critical role in the regulation of the inflammatory response and thus in atherosclerosis [41]. Elevated miR155 levels have been associated with inflammatory macrophages and atherosclerotic lesions, but its effects may differ in early and advanced atherosclerosis depending on plaque content [42,43]. Nonetheless, inhibition of miR155 in immune cells appears to be a promising strategy to limit the development of atherosclerosis [44], certain cancers [45], and inflammatory diseases such as rheumatoid arthritis [46]. This study, with this original strategy, reinforces the promise of targeting miR155 to limit VC in peripheral atherosclerotic arteries, but additional experiments with miR155 targeting already diseased and calcified arteries need to assess its potential clinical value. To further explore the clinical impact of miR155 on calcification, which was not determined in the present study, future experiments will need to investigate its impact on early- and late-stage calcification through local or systemic approaches.

## 4. Materials and Methods

### 4.1. Biological Samples

The ECLAGEN biocollection includes human carotid bifurcation and common femoral artery samples. Details of these biocollections have been described in previous publications [47,48]. Pathological atheromatous tissues were collected during endarterectomy, and healthy arteries were collected from organ donors. In the operating room, the atheromatous lesions obtained via endarterectomy were cut into two pieces, one for RNA extraction and molecular analysis and the other for histological analysis. Stratification of lesion calcification was performed after histomorphometry analysis of mineralized structures, as detailed in our previous studies [11]. The collection of human samples was carried out according to strict ethical standards. Written informed consent was obtained from each patient included in the study (research protocol #PFS09-014, authorized on 23 December 2009 by the French “Agence de Biomédecine”). For deceased donors, no opposition to organ donation was checked and written consent from the donor’s family was obtained, and therapeutic arterial samples were always prioritized. The study protocol conformed to the ethical guidelines of the 1975 Declaration of Helsinki, and the study protocol was priorly approved by the Institution’s ethics committee on research on humans. The legal authorizations were obtained from the French Ministry of Research (n DC-2008-402), the National Commission for Informatics and Freedoms (CNIL, n 1,520,735 v 0), and the Local Ethics Committee (GNEDS, Groupe Nantais d’Ethique dans le Domaine de la Santé). Histological study of this biocollection had been carried out in previous studies [11,26].

### 4.2. RNA Extraction, miR Microfluidic Arrays, and Transcriptomic Analysis

Expression of 753 human miRs was carried out with microfluidic cards (TaqMan™ Array Human MicroRNA cards, ThermoFisher, Waltham, MA, USA). The analysis was conducted on 60 arterial samples from the ECLAGEN biocollection: 20 atherosclerotic carotid arteries (10 calcified, 10 none calcified), 20 atherosclerotic common femoral arteries (10 ossified, 10 none ossified), 10 healthy carotid arteries, and 10 healthy common femoral arteries. Adventitia was removed before RNA extraction. Total RNA extraction from tissue and primary cells was performed using the Rneasy Plus Micro kit, following the manufacturer’s instructions (Qiagen, Venlo, The Netherlands). For tissue samples, Tissuelyzer II (Qiagen) was used for homogenization before RNA extraction. Following the completion of the Qiazol and column-based extraction procedure, RNA concentration was determined by spectrophotometry (NanoDrop 1000, ThermoFisher). Reverse transcription was achieved with the RT Maxima H Minus Enzyme Mix kit according to the manufacturer’s recommendations (ThermoFischer). mRNA analysis was performed on microarray data from the same samples as previously reported, deposited in NCBI’s Gene Expression Omnibus (GSE100927). Gene expression differences between calcified and non-calcified atherosclerotic carotid arteries and between ossified and non-ossified femoral arteries were analyzed using the limma R/Bioconductor software package (v 3.10) [49]. The obtained results were visualized using the VolcaNoseR web app (v1.0.1) [50].

### 4.3. Cell Culture and Mineralization

Primary human aortic smooth muscle cells were isolated from human aorta (PromoCell, Heidelberg, Germany, 3 separate donors) and cultured in a dedicated proliferation medium (Smooth Muscle Cell Growth Medium 2, PromoCell), including fetal calf serum (5%), growth factors (Epidermal Growth Factor and Basic Fibroblastic Growth Factor), and insulin (5 µg/mL). Penicillin and streptomycin (PS) were added to the 1% concentration (Sigma Aldrich, St. Louis, MO, USA). VSMCs were cultured in two batches of pro-mineralizing Dulbecco’s Modified Eagle’s Medium (DMEM), both containing high glucose concentrations (Gibco, Paisley, Scotland; Thermofisher). We added fetal calf serum (3%) and inorganic phosphate at a concentration of 3 mM (“Pi-enriched Medium”) by adding NaH_2_PO_4_ (Sigma Aldrich) and Na_2_HPO_4_ (Sigma Aldrich) solutions (4:1 ratio). For the second medium (“OB medium”), we added fetal calf serum (10%), Dexamethasone 10^−7^ M, and vitamin D3 10^−8^ M (from day 4 to day 21) and β-Glycerophosphate 10 mM and ascorbic acid 0.25 mM (from day 8 to day 21). Cells were then fixed with absolute alcohol for 20 min at 4 °C. Mineralization staining was performed with Rouge Alizarine (Sigma Aldrich) for 20 min at room temperature. Stained wells were individually photographed with a binocular magnifying glass (Zeiss Stemi 200-C, Göttingen, Germany and AxioVision Rel 4.8 software). Mineralization was quantified with Image Pro Plus software (v6.0) by calculating the ratio of the mineralized area (stained in red) to the total well area.

### 4.4. Cell Transfection and Functional Validation of miRs of Interest

miR overexpression with mimic miRs (30 nM, Qiagen) was achieved by lipofection (Lipofectamine RNAiMax, ThermoFischer). For mineralization experiments, the pro-mineralizing medium was added 24 h after transfection, and this was repeated every 7 days.

### 4.5. Gene and miR Expression Analysis

Transcriptional analysis was performed by Real-Time qPCR on the CFX96 (Bio-Rad, San Francisco, CA, USA) detection device with Power Sybr green PCR Master Mix (ThermoFisher). Target gene expression was normalized to *GAPDH* expression, and the comparative cycle threshold (Ct) method was used to calculate the relative expression of target mRNAs. Primers of target mRNAs were as follows: *Glyceraldehyde-3-Phosphate Dehydrogenase (GAPDH)* R–GGTGCAGGAGGCATTGCT/F–TGGGTGTGAACCATGAGAAGTATG, *Runt-Related Transcription Factor 2 (RUNX2)* R–GCTCTTCTTACTGAGAGTGGAAGG/F–GCCTAGGCGCATTTCAGA, *Osteocalcin (OCN)* R–GTGGTCAGCCAACTCGTCA/F–GGCGCTACCTGTATCAATGG, *Osterix (OSX)* R–GCCTTGCCATACACCTTGC/F–CTCCTGCGACTGCCCTAAT, *Osteopontin (OPN)* R–CAATTCTCATGGTAGTGAGTTTTCC/F–GAGGGCTTGGTTGTCAGC, *Collagen Alpha-1(I) Chain (COL1A1)* R–GCTCCAGCCTCTCCATCTTT/F–CTGGACCTAAAGGTGCTGCT, *Alkaline Phosphatase (ALP)* R–GGTCACAATGCCCACAGATT/F–AACACCACCCAGGGGAAC, *Bone Sialo Protein (BSP)* R–CAGTCTTCATTTTGGTGATTGC/F–CAATCTGTGCCACTCACTGC, *Aggrecan (ACAN)* R–GACACACGGCTCCACTTGAT/F–CCCCTGCTATTTCATCGACCC, *SRY-Box Transcription Factor 9 (SOX9)* R–TCGCTCTCGTTCAGAAGTCTC/F-GTACCCGCACTTGCACAAC, *Actin Alpha 2 (ACTA2)* R–CCGGCTTCATCGTATTCCTGTT/F–TCCTTCATCGGGATGGAGTCT, *Tropomyosin 1 (TPM1)* R–CTCCTCTGCACGTTCCAGGT/F–AGGAGCGTCTGGCAACAGCT, *Transgelin (SM22-Alpha)* R–CACCAGCTTGCTCAGAATCA/F–CAGTGTGGCCCTGATGTG, *Calponin (CNN1)* R–GTACTTCACTCCCACGTTCACCTT/F–GAACATCGGCAACTTCATCAAGGC, and *CD73* (QuantiTect N° QT00027279, Qiagen). We analyzed miR expression by RT-qPCR using the miRCURY LNA RT kit and the miRCURY qPCR LNA kit (Qiagen). The miR primers were provided by Qiagen (LNA miRNA PCR Assay, Qiagen). The expression of miRs was normalized to miR SNORD 44 and 48 expression.

### 4.6. In Silico Analysis of the miR Candidates

We determined the associated genes of the miR candidates using the miRWalk 2.0, miRbase, and miRBD databases [51] and the main associated biological functions using the REVIGO online database accessed on 1 June 2023 (https://github.com/rajko-horvat/RevigoWeb) and the gene ontology resource accessed on 1 June 2023 (http://geneontology.org/). We performed a cross-analysis of the data based on the transcriptomic analysis of the ECLAGEN biocollection already published [11,26] with the selected miRs and the gene ontology resource accessed on 1 June 2023 (http://geneontology.org/). We selected target genes of the miRs of interest found to be associated with calcified plaques and involved in the process of vascular calcification/ossification and/or cell mineralization.

### 4.7. Protein Expression Analysis

Protein extraction was performed with RIPA lysis buffer containing protease inhibitors (P8340, Sigma Aldrich^®^). Quantification was conducted by spectrophotometric analysis using the BCA Protein Assay kit (ThermoFisher). Protein expression analysis was performed by Western blotting after migration on Acrylamide 4–15% gradient gel with Bis-Tris buffer (Bio-Rad^®^). The transfer was then realized on a nitrocellulose membrane TransBlot Turbo (Bio-Rad^®^). We used the following antibodies: anti-CD73 (ab 133582 Abcam, Waltham, MA, USA) and anti-Vinculin (ab 129002 Abcam). We used chemiluminescence as a detection method with the Clarity Western ECL Substrate (Bio-Rad^®^) reagent.

### 4.8. Statistical Analysis

Statistical analysis and graphics were performed and produced using GraphPad Prism^®^ software (v8) (GraphPad Software^®^, Inc., La Jolla, CA, USA). For comparison between two groups, an unpaired Student’s t-test was performed. For comparison among three or more conditions, non-parametric one-way ANOVA (Mann–Whitney test) followed by Dunnett’s post-test was performed. A *p*-value of less than 0.05 was considered significant. For in vitro analysis, all statistical tests were based on data from at least *n* = 3 independent experiments (experiments using independent cell batches).

## Figures and Tables

**Figure 1 ijms-26-09349-f001:**
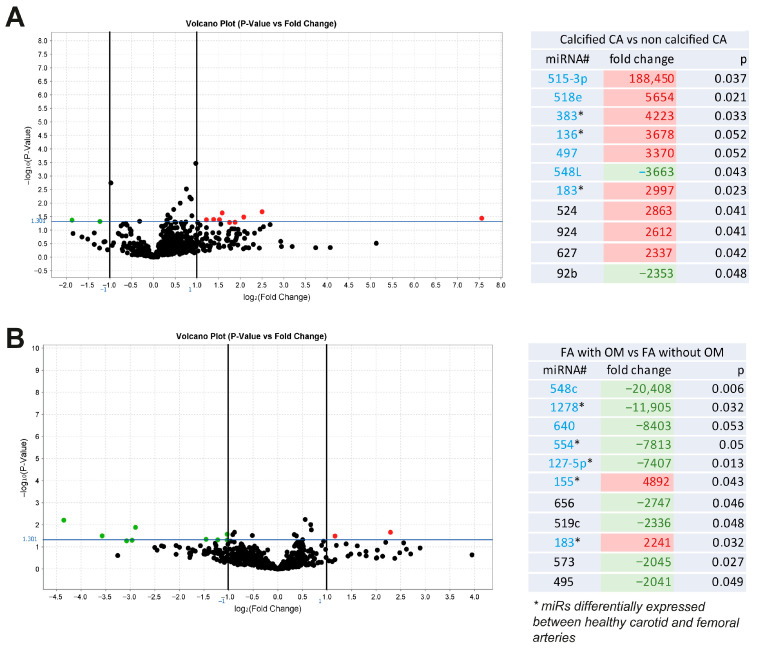
miRs associated with VC and ossification. Volcano plot graphs and tables representing fold changes of miRs differentially expressed between calcified and non-calcified pathological carotid arteries (CA) (*n* = 20 patients) (**A**) and between ossified and non-ossified femoral arteries (FA) (*n* = 20 patients) (**B**). CA = Carotid Arteries; FA = Femoral Arteries; OM = Osteoid Metaplasia.

**Figure 2 ijms-26-09349-f002:**
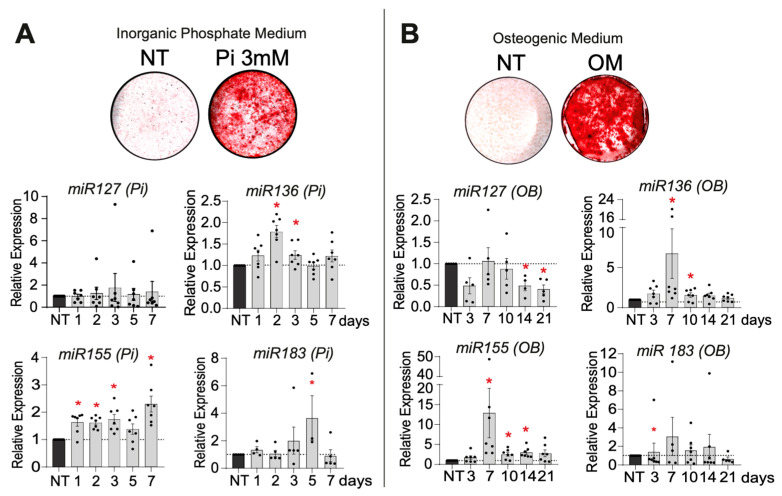
Expression and regulation of miRs during VSMC mineralization in vitro. miR expression levels during VSMC mineralization induced by inorganic phosphate (3 mM) (**A**) and osteogenic medium (**B**) for each miR candidate: 127, 136, 155, and 183. Bars represent means ± SEMs (* *p* < 0.05). Pi = Inorganic Phosphate; OB = Osteogenic Medium; NT = Non-Treated.

**Figure 3 ijms-26-09349-f003:**
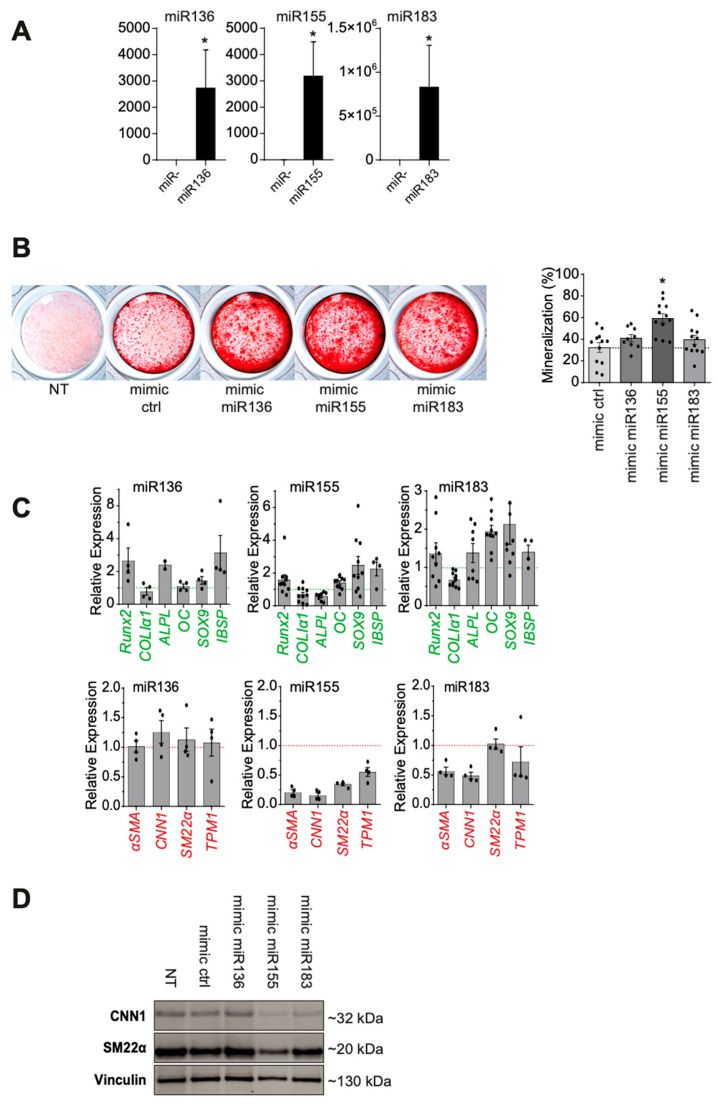
Functional impact of miR candidates on human VSMC mineralization. (**A**) miR136, -155, and -183 overexpression in VSMCs 7 days after transfection of individual miR mimics. (**B**) Representative images of Alizarin red staining for mineralization (in red) of VSMCs 7 days after inducing miR overexpression (negative control, mir136, miR155, and miR183 mimics) in inorganic phosphate-enriched medium (3 mM) and quantification of the mineralized area percentage (stained area/total area) between conditions. (**C**) Transcriptional regulation of osteoblastic (in green) and contractile-associated genes (in red) 7 days after inducing miR overexpression compared to a mimic negative control. Bars represent means ± SEMs (* *p* < 0.05). (**D**) Representative Western blotting for contractile-associated proteins 7 days after miR mimic transfection. NT = Non-Treated.

**Figure 4 ijms-26-09349-f004:**
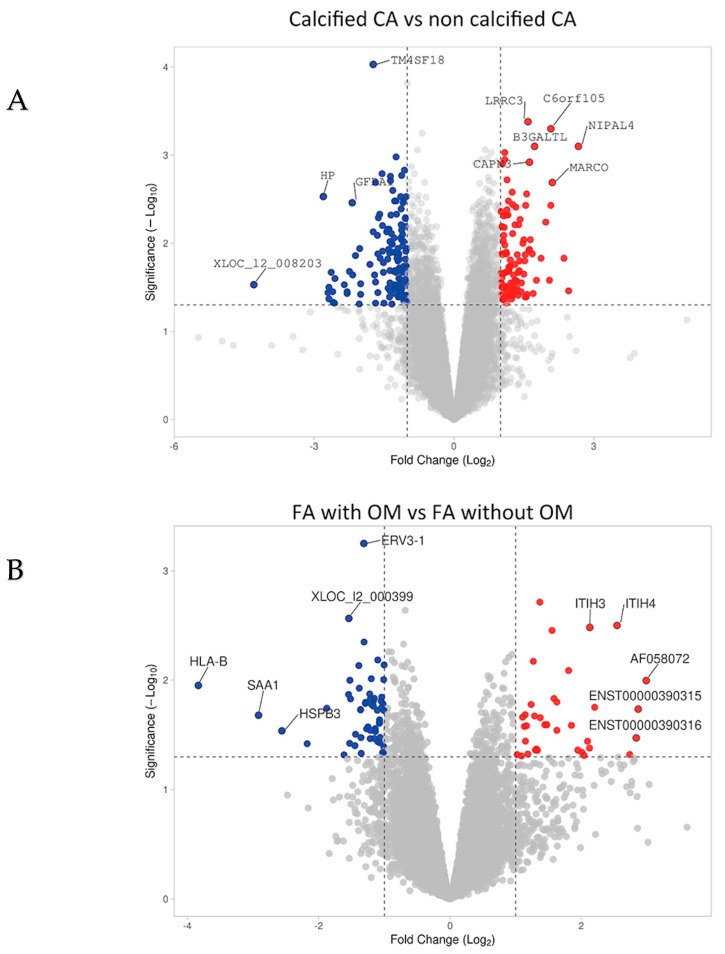
Genes associated with VC. Volcano plot graphs representing fold changes of genes differentially expressed between calcified and non calcified pathological carotid arteries (**A**) and between ossified and non-ossified pathological femoral arteries (*n*= 20) (**B**). (**C**) REVIGO scatterplot of their gene ontology annotations (biological functions) highlighting their contributing roles in the biomineralization process. CA = Carotid Arteries; FA = Femoral Arteries; OM = Osteoid Metaplasia.

**Figure 5 ijms-26-09349-f005:**
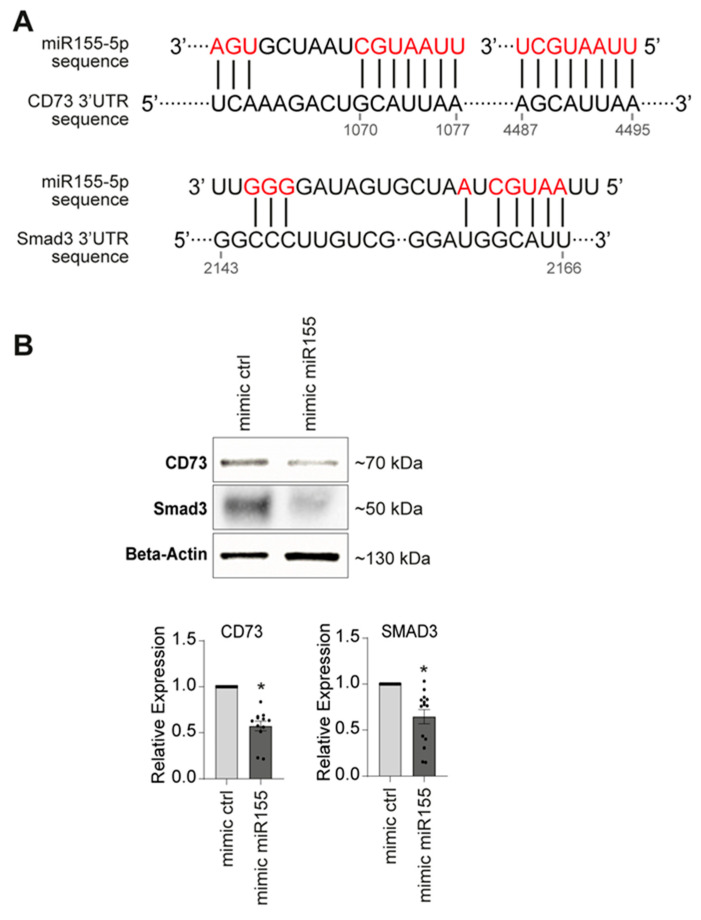
Putative targets for miR155. (**A**) Putative interaction sites between miR155 and *Smad3* (NCBI gene ID: 4088) and *CD73* (NCBI gene ID: 5167). (**B**) Smad3 and CD73 protein and mRNA levels 2 days following overexpression of indicated miR mimics. Bars represent means ± SEMs (* *p* < 0.05).

## Data Availability

miRNA analysis was performed on microarray data deposited in the NCBI’s Gene Expression Omnibus (GSE100927). Other data are available upon request from the authors.

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
