# Peer review of "Identification of miR136, miR155, and miR183 in Vascular Calcification in Human Peripheral Arteries"

_ijms, 2025, doi:10.3390/ijms26199349_

Round 1

Reviewer 1 Report

Comments and Suggestions for Authors

This is a clinically relevant study that identifies novel microRNAs associated with vascular calcification (VC) in human peripheral arteries by the analysis of peripheral arterial samples and in vitro models of vascular smooth muscle cell (VSMC) mineralization. The study combines miRNomic and transcriptomic profiling to detect specific miRNAs as potential therapeutic targets in VC. It demonstrates the functional significance of miR155 in promoting VSMC mineralization and identifies target genes. The manuscript is well-structured and addresses an important clinical problem, as VC is a critical factor of cardiovascular morbidity and mortality. The introduction section of the manuscript gives an adequate background about the topic.‎ The ‎methods are detailed and reliable. The results and figures are clearly presented. The ‎discussion analyzes the work and has sufficient information about prior studies.‎The conclusions are logical. The references are adequate. However, several points need to be addressed before the manuscript can be considered for publication:

  1. The novelty is limited, as previous studies have already associated miR155 with vascular calcification and atherosclerosis. The manuscript should more clearly delineate how its findings advance knowledge beyond current literature.
  2. Line 387-388: "Dulbecco’s Modified Eagle’s Medium" is not αMEM. This should be corrected.
  3. The results section is difficult to follow. The shift from the initial selection of 12 miRs to the focus on only four miRs 127, 136, 155, and 183, and excluding the other eight miRs, needs clearer justification.
  4. The study depends on overexpression experiments with miR mimics. The absence of loss-of-function methodologies utilizing antagomirs limits the mechanistic conclusions. Inclusion of such data, or acknowledgment of this limitation, can strengthen the study.
  5. The study demonstrates the downregulation of CD73 and Smad3 by miR155; however, direct binding was not validated by luciferase reporter assays for the 3’ UTR. This weakens the evidence for a direct mechanical link.
  6. The data for miR-127 shows significant downregulation in the OB model (Fig. 2B) and no effect on mineralization when overexpressed (Fig. 3B). This suggests it may function as a suppressor. This is a noteworthy finding that is under-discussed.
  7. The manuscript insufficiently acknowledges its limitations, such as the absence of in vivo validation or animal models.

Author Response

We would like to thank the reviewers for their constructive comments and suggestions for improving our manuscript. We have addressed each of their remarks in the following section. We hope these responses will support the article's consideration for publication in the International Journal of Molecular Sciences.

Reviewer 1:

This is a clinically relevant study that identifies novel microRNAs associated with vascular calcification (VC) in human peripheral arteries by the analysis of peripheral arterial samples and in vitro models of vascular smooth muscle cell (VSMC) mineralization. The study combines miRNomic and transcriptomic profiling to detect specific miRNAs as potential therapeutic targets in VC. It demonstrates the functional significance of miR155 in promoting VSMC mineralization and identifies target genes. The manuscript is well-structured and addresses an important clinical problem, as VC is a critical factor of cardiovascular morbidity and mortality. The introduction section of the manuscript gives an adequate background about the topic.‎ The ‎methods are detailed and reliable. The results and figures are clearly presented. The ‎discussion analyzes the work and has sufficient information about prior studies.‎ The conclusions are logical. The references are adequate. However, several points need to be addressed before the manuscript can be considered for publication:

  1. The novelty is limited, as previous studies have already associated miR155 with vascular calcification and atherosclerosis. The manuscript should more clearly delineate how its findings advance knowledge beyond current literature.

We thank the reviewer for the comment and suggestion. We have amended the manuscript to clarify its originality and how our study implement the current literature.

“To our knowledge, this study is the first aimed to identify miRs associated with VC in a cohort of human atherosclerotic peripheral lesions with its putative targets using cross-analysis with transcriptomic data, and our results highlight CD73 and Smad3 as likely direct or indirect targets that contributes to VC both in calcified peripheral arteries and in human primary mineralizing SMCs.” (Line 340-344)

“This study, with this original strategy, reinforces the promises in targeting miR155 for limiting VC in peripheral atherosclerotic arteries, but additional experiments with miR155 targeting already diseased and calcified arteries need to assess its potential clinical value. To further explore the clinical impact of miR155 on calcification, which is not included in the present study's, future experiments will need to investigate its impact on early- and late-stage calcification, through local or systemic approaches. (Line 350-356)

  1. Line 387-388: "Dulbecco’s Modified Eagle’s Medium" is not αMEM. This should be corrected.

We apologize for the confusion; this has been corrected.

“VSMC were cultured in two pro-mineralizing Dulbecco’s Modified Eagle’s Medium (DMEM) containing both high glucose (Gibco, Thermofisher)” (Line 402)

  1. The results section is difficult to follow. The shift from the initial selection of 12 miRs to the focus on only four miRs 127, 136, 155, and 183, and excluding the other eight miRs, needs clearer justification.

We thank the reviewer for pointing out this lack of clarity. We have amended the manuscript to more clearly explain our strategy of focusing on potential miRs that are directly implicated in smooth muscle cell mineralization. Of the 12 miRs identified, only four were actually expressed in VSMCs in vitro. While we agree that the remaining eight are likely important as well, they are likely to be expressed or regulated by other cell types in calcified lesions, such as macrophages/immune cells, endothelial cells, pericytes, and fibroblasts/myofibroblasts. This question is part of the study's future directions.

“2.2. Selection of the most promising miRs regulated during VSMC mineralization

VSMCs in lesions are believed to be the main drivers of VC because these cells can acquire osteoblastic properties. Our strategy focused on potential miRs directly implicated in VSMC mineralization, considering that some of the miRs identified in the entire lesion could also be indirectly related to calcification. To select the best candidates most likely to regulate VSMC mineralization, we analyzed the relative expression levels of the 12 miRs of interest in primary arterial VSMCs to identify those expressed in VSMCs and regulated during mineralization.” (Line 122-129).

2.2.2. Selection of the miR candidates according to their expression and regulation during VSMC mineralization.

During Pi-driven mineralization, we observed some regulations for 3 miR candidates (miR136, 155, and 183). An increased expression of miR136 (1.828 fold at day 2), miR155 (2.227 fold at day 7), and miR183 (4.53 fold at day 5) was apparent (Fig.2-A). The OB pro-mineralizing medium also regulated transiently 4 miRs of interest. We observed an up-regulation of miR136 (2.346-fold at day 7), miR155 (5.242-fold at day 7), and miR183 (3.45-fold at day 7). Conversely, we observed a down-regulation of miR127 (0.357 at day 21) (Fig.2-B). Among the 12 mRs of interest screened, miR383, miR497, miR515, miR518, miR548, miR554, miR640, and miR1278 were not retained as miRs candidates, as their relative expression levels were undetectable by qPCR during mineralization induced by both pro-mineralizing conditions. (Line 154-165).

  1. The study depends on overexpression experiments with miR mimics. The absence of loss-of-function methodologies utilizing antagomirs limits the mechanistic conclusions. Inclusion of such data, or acknowledgment of this limitation, can strengthen the study.

We agree with the reviewer that this is part of the limitations of the study. We did perform multiple anti-miR experiments with miR inhibitors (Thermofisher), but we consistently failed to see any significant decrease in miR expression following transfection of our primary human VSMCs, making difficult any interpretation of functional mineralization assays. We acknowledged this limitation in the discussion section.

While we couldn’t validate miR155 inhibition with transient transfection experiments, using murine aorta rings, miR155 deficiency attenuated calcification, and miR155-/- mice showed reduced VC induced by vitamin D3 [37]. (Line 323-324)

  1. The study demonstrates the downregulation of CD73 and Smad3 by miR155; however, direct binding was not validated by luciferase reporter assays for the 3’ UTR. This weakens the evidence for a direct mechanical link.

We thank the reviewer for the comment and agree that this is a limitation of the study. While direct regulation is certainly possible, we could not rule out the possibility of an indirect regulation, similar to how miR127 appears to induce CD73 indirectly. Nevertheless, we believe that this association remains highly relevant in the context of tissue mineralization and osteogenic differentiation. We have made the following amendments to the manuscript to clarify this point.

 “As miR127 was also shown to increase CD73 expression, this further confirm the key role of CD73 in SMC mineralization and tissue calcification [40]. Additional exploratory and mechanistic studies will be required to establish the precise direct and/or indirect link between miR155 and miR127 on CD73 and Smad3 in human calcified tissues”. (Line 336-339)

  1. The data for miR-127 shows significant downregulation in the OB model (Fig. 2B) and no effect on mineralization when overexpressed (Fig. 3B). This suggests it may function as a suppressor. This is a noteworthy finding that is under-discussed.

We appreciate the reviewer's feedback and have amended the discussion to incorporate this pertinent suggestion.

“Furthermore, we observed the decrease in miR127 expression during mineralization induced by an osteogenic milieu containing dexamethasone, vitamin D3, β-glycerophosphate, and ascorbic acid. Although miR127 overexpression did not affect Pi-induced mineralization, miR127 might suppress VSMC osteoblastic differentiation because specific mechanisms can regulate each of these mineralization models [36]. As miR127 was also shown to increase CD73 expression, this further confirm the key role of CD73 in VSMC mineralization and tissue calcification [40]” (Line 331-339)

  1. The manuscript insufficiently acknowledges its limitations, such as the absence of in vivo validation or animal models.

We agree with the reviewer and acknowledged more clearly the limitations of this study.

While we couldn’t validate miR155 inhibition with transient anti-miR transfection experiments, using murine aorta rings, miR155 deficiency attenuated calcification, and miR155-/- mice showed reduced VC induced by vitamin D3 [37]. (Line 323-324)

“Additional exploratory and mechanistic studies will be required to establish the precise direct and/or indirect link between miR155 and miR127 on CD73 and Smad3 in human calcified tissues.” (Line 337-339)

“To further explore the clinical impact of miR155 on calcification, which is not included in the present study's, future experiments will need to investigate its impact on early- and late-stage calcification, through local or systemic approaches.” (Line 353-356)

Reviewer 2 Report

Comments and Suggestions for Authors

Vascular calcification (VC) is a severe pathological event in which minerals are deposited in the vessel muscular wall and is associated with an increased risk of stroke and heart attack. In the present study, Le Corvec et al. used calcified and non-calcified human atherosclerotic arteries as well as human vascular smooth muscle cells (VSMC) to identify microRNAs (miRs), their target molecules, and their role in cell mineralization. Among the 12 miRs associated with smooth muscle cells mineralization, only three miRs showed some regulation during differentiation of VSMCs into an osteoblast-like phenotype. miR155 proved to be the most effective candidate in VSMC mineralization and its overexpression promoted the upregulation of several osteoblastic genes. In addition, the authors could demonstrate a connection between miR155 and two regulatory pathways (Smad3 and CD73), which are involved in arterial calcification. The authors concluded that inhibition of miR155 could attenuate VC in peripheral atherosclerotic arteries.

This is a well-written and clearly presented study, to which I have only a few minor comments.

Line (L) 72-74: Perhaps the authors could add that the plaques in the femoral arteries also exhibit bone-like histology [see Herisson et al. Atherosclerosis. juin 2011;216(2):348-54].

L 178: Explain CML

L 337: ...these miRs... – which miRs are the authors referring to here?

For clarity, the authors should explain the abbreviations used in the figures. For example, CA = carotid artery and FM = femoral artery. “OM” in Fig. 1B and 4B - what does that mean?

“NT” in Fig. 2 – what does it mean?

Author Response

We would like to thank the reviewers for their constructive comments and suggestions for improving our manuscript. We have addressed each of their remarks in the following section. We hope these responses will support the article's consideration for publication in the International Journal of Molecular Sciences.

Reviewer 2:

Vascular calcification (VC) is a severe pathological event in which minerals are deposited in the vessel muscular wall and is associated with an increased risk of stroke and heart attack. In the present study, Le Corvec et al. used calcified and non-calcified human atherosclerotic arteries as well as human vascular smooth muscle cells (VSMC) to identify microRNAs (miRs), their target molecules, and their role in cell mineralization. Among the 12 miRs associated with smooth muscle cells mineralization, only three miRs showed some regulation during differentiation of VSMCs into an osteoblast-like phenotype. miR155 proved to be the most effective candidate in VSMC mineralization and its overexpression promoted the upregulation of several osteoblastic genes. In addition, the authors could demonstrate a connection between miR155 and two regulatory pathways (Smad3 and CD73), which are involved in arterial calcification. The authors concluded that inhibition of miR155 could attenuate VC in peripheral atherosclerotic arteries.

This is a well-written and clearly presented study, to which I have only a few minor comments.

  1. Line (L) 72-74: Perhaps the authors could add that the plaques in the femoral arteries also exhibit bone-like histology [see Herisson et al. Atherosclerosis. juin 2011;216(2):348-54].

We thank the reviewer for this suggestion. We have amended the manuscript accordingly.  

“Histological analysis showed mostly amorphous calcification in carotid lesions while plaques in femoral arteries exhibit a higher calcium content and are also more prone to bone-like calcification and osteoïd metaplasia [24].” (Line 71-74)

  1. L 178:Explain CML

We apologize for using the French acronym for "SMC." We have corrected the manuscript.

« We also observed a significant decrease in the expression of the VSMC-specific markers α-SMA… » (Line 188)

  1. L 337:...these miRs... – which miRs are the authors referring to here?

We meant miR155 mostly, as miR127, miR136 and miR183 had no clear functional impact on VSMC mineralization.

“To further explore the clinical impact of miR155 on calcification, which is not included in the present study's, future experiments will need to investigate its impact on early- and late-stage calcification, through local or systemic approaches.” (Line 353-356)

  1. For clarity, the authors should explain the abbreviations used in the figures. For example, CA = carotid artery and FM = femoral artery. “OM” in Fig. 1B and 4B - what does that mean? “NT” in Fig. 2 – what does it mean?

We apologize for the lack of clarity. The meaning of these abbreviations has been included in the revised figure legends (Figure 1. Line 120-121; Figure 2. Line 169-170; Figure 3. Line 213; Figure 4. 249).

Round 2

Reviewer 1 Report

Comments and Suggestions for Authors

I would like to thank the authors for considering my comments and for the revisions they made to the manuscript. The authors have provided a detailed point-by-point response and have made significant improvements to the manuscript. My major concerns have been adequately addressed.